# Structure and antigenicity of divergent Henipavirus fusion glycoproteins

Ariel Isaacs [1,4], Yu Shang Low[1,4], Kyle L. Macauslane [1], Joy Seitanidou [1], Cassandra L. Pegg[1], Stacey T. M. Cheung[1], Benjamin Liang [1], Connor A. P. Scott[1], Michael J. Landsberg[1,2], Benjamin L. Schulz [1,2], Keith J. Chappell [1,2,3], Naphak Modhiran [1,5] ✉ & Daniel Watterson [1,2,5] ✉

In August 2022, a novel henipavirus (HNV) named Langya virus (LayV) was isolated from patients with severe pneumonic disease in China. This virus is closely related to Mòjiāng virus (MojV), and both are divergent from the bat-borne HNV members, Nipah (NiV) and Hendra (HeV) viruses. The spillover of LayV is the first instance of a HNV zoonosis to humans outside of NiV and HeV, highlighting the continuing threat this genus poses to human health. In this work, we determine the prefusion structures of MojV and LayV F proteins via cryogenic electron microscopy to 2.66 and 3.37 Å, respectively. We show that despite sequence divergence from NiV, the F proteins adopt an overall similar structure but are antigenically distinct as they do not react to known antibodies or sera. Glycoproteomic analysis revealed that while LayV F is less glycosylated than NiV F, it contains a glycan that shields a site of vulnerability previously identified for NiV. These findings explain the distinct antigenic profile of LayV and MojV F, despite the extent to which they are otherwise structurally similar to NiV. Our results carry implications for broad-spectrum HNV vaccines and therapeutics, and indicate an antigenic, yet not structural, divergence from prototypical HNVs.

Henipaviruses (HNVs) are regarded as the most lethal paramyxoviruses, with a case fatality rate of ~70%[1,2]. The HNV genus was first established after identification of the two prototypical members: Nipah virus (NiV) and Hendra virus (HeV). Since their emergence in the 1990's, numerous spillover events of HeV from bats to horses have occurred in Australia, resulting in several human exposures[3]. Furthermore, NiV spillovers to humans occur almost annually in Bangladesh, and several outbreaks have also been recorded in India and the Philippines[3]. In recent years, the HNV genus has expanded significantly, with new viruses discovered in Africa, Australia, Asia and South America. These include Cedar virus (CedV), isolated from flying foxes in Australia[4]; Ghana virus (GhV), isolated from bats in Africa[5];

Mòjiāng virus (MojV), sequenced from rats in China[6]; Gamak & Daeryong viruses, discovered in shrews in the Republic of Korea[7], and most recently Langya virus (LayV), identified in a throat swab from a human patient in China[8]. Outside of NiV and HeV, LayV is the only other HNV that is known to infect humans, however it is suspected that MojV and GhV also possess pathogenic potential.

Henipaviral diseases are listed on the WHO list of priority diseases that require exigent research into vaccines and therapeutics. This is largely due to the high associated case fatality rate and the global migratory patterns of the *Pteropus* fruit bats, which act as the main animal reservoir for HNVs[9]. With the identification of MojV and LayV, the animal reservoir of HNVs has expanded to rats and shrews.

[1]School of Chemistry and Molecular Biosciences, The University of Queensland, Brisbane, Australia. [2]Australian Infectious Disease Research Centre, The University of Queensland, Brisbane, Australia. [3]Australian Institute for Bioengineering and Nanotechnology, Brisbane, Australia. [4]These authors contributed equally: Ariel Isaacs, Yu Shang Low. [5]These authors jointly supervised this work: Naphak Modhiran, Daniel Watterson. ✉e-mail: n.modhiran@uq.edu.au; d.watterson@uq.edu.au

Together, these factors increase the likelihood of spillover to livestock and humans. There are currently no approved vaccines for HNVs, however a HeV G vaccine is available for veterinary use and an analogous subunit vaccine is currently in clinical development for human use[10–13]. Based on sequence homology between LayV and HeV (~24%), this vaccine candidate is unlikely to confer protection to divergent HNVs. As such, there is a clear need for improved vaccine preparedness against these emerging pathogens.

An alternate vaccine target to HNV G is the fusion (F) glycoprotein. The HNV F protein is a trimeric type I fusion protein that is initially expressed as an F precursor ($F_0$), consisting of three domains (DI, DII and DIII), a C-terminal stem, a transmembrane domain and a cytoplasmic tail. During infection, the $F_0$ protein is cleaved by cathepsin L protease into $F_1$ and $F_2$ subunits, which are linked by disulfide bonds and together constitute the fusogenic F protein[14–16]. The prefusion conformation of NiV and HeV F have been determined along with the fusion core of NiV[17–19]. These structures highlight the large conformational changes that take place within F during fusion, where two heptad repeats in DIII and the stem (HRA and HRB) coalesce into a six-helix bundle, which is required for insertion of the fusion peptide into the host cell membrane[20]. This outlines a rational basis for neutralizing antibody discovery and structure-based vaccine design against prefusion F. Indeed, several antibodies targeting the prefusion conformation of F have been isolated and shown to be neutralizing and protective[21–24]. Moreover, vaccine studies with prefusion-stabilized NiV F have been shown to elicit potent neutralizing responses[25–28]. While these studies yielded vital information into HNV vaccine design, it is unknown whether this immunity extends to other divergent HNVs such as MojV and LayV.

The F and G proteins of HNV are glycoproteins, as they are post-translationally modified with oligosaccharide structures known as glycans. Two common types of glycans, *N*-linked or *O*-linked, are covalently attached to nitrogen or oxygen atoms in amino acid side chains, respectively. Protein glycosylation can alter immunity and antibody sensitivity by shielding or exposing viral protein epitopes, as has been observed for SARS-CoV-2[29] and HIV[30]. Glycosylation can also impact the structures and dynamics of proteins. With respect to HNV F proteins, mutations of the sites of *N*-glycosylation have shown they are required for proper folding and processing of NiV and HeV F[31,32]. Site-specific glycosylation of glycoproteins can be studied using mass spectrometry glycoproteomics, as has been applied to HeV G[31]. However, there are few studies investigating the glycosylation of HNV F proteins, which is likely to be important in the context of immunity and vaccine design.

Structure-based rational vaccine design has helped improve and implement several vaccine candidates, including the SARS-CoV-2 prefusion-stabilized two-proline spike antigen formulated in Pfizer and Moderna mRNA vaccines and respiratory syncytial virus (RSV) F now in late-stage clinical development[33–37]. These antigens are stabilized in the prefusion conformation by a set of mutations and/or a trimerization domain, allowing for elicitation of prefusion-specific antibodies. A similar approach to that undertaken with RSV was recently shown to stabilize the NiV and HeV F glycoproteins in the prefusion conformation[26]. In parallel, we have previously shown that prefusion NiV F can be stabilized by a novel molecular clamp trimerization domain in the absence of heterologous mutations[25,28]. Here, we use a similar approach to determine the structures of LayV and MojV F glycoproteins in the prefusion conformation via cryogenic electron microscopy (cryo-EM) and to characterize their glycosylation profiles with mass spectrometry glycoproteomics, in order to inform future vaccine design and therapy against these emerging viruses.

## Results

### Cryo-EM structures of LayV and MojV reveal a similar architecture to NiV F despite sequence divergence

Based on the sequence of LayV, published in August 2022, we expressed the F ectodomain residues 1-478 without any heterologous stabilizing mutations (Fig. 1a)[8]. To stabilize the prefusion form, we added a proprietary molecular clamp (clamp2) domain, analogous to that previously described[25,28,38], to the C-terminus of F. This construct

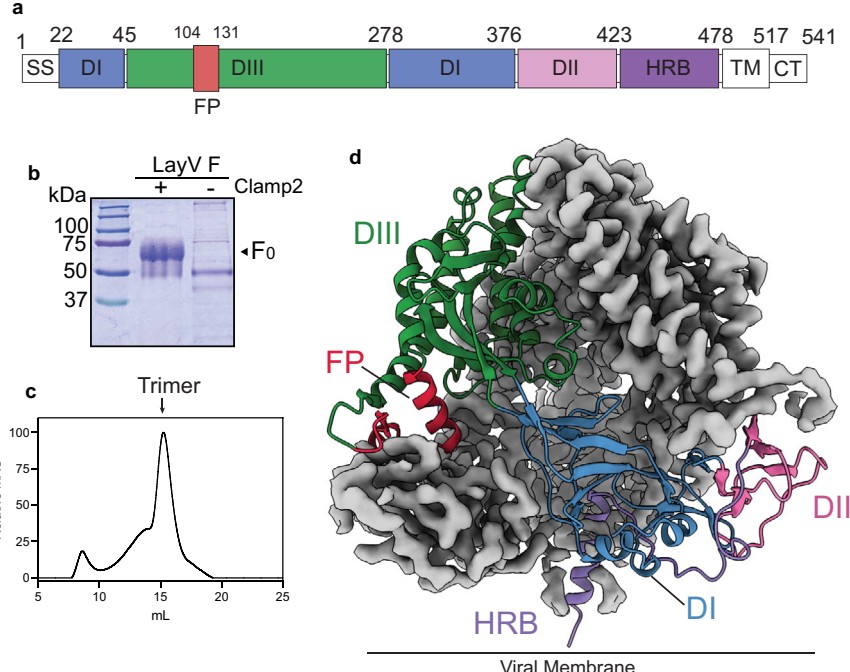

**Fig. 1 | Cryo-EM structure of LayV F glycoprotein in the prefusion form. a** Gene schematic of LayV F with each domain coloured. Domains coloured in white are not included in the final structure. **b** SDS-PAGE of purified LayV F proteins under reducing conditions either stabilized with clamp2 or unstabilised. **c** SEC of LayV F clamp2 ran on Superose 6 Increase 10/300 GL column. Trimer peak is indicated. **d** Cryo-EM map (grey) resolved to 3.67 Å and generated model coloured as in (**a**) of prefusion LayV F. Source data are provided as a Source Data file.

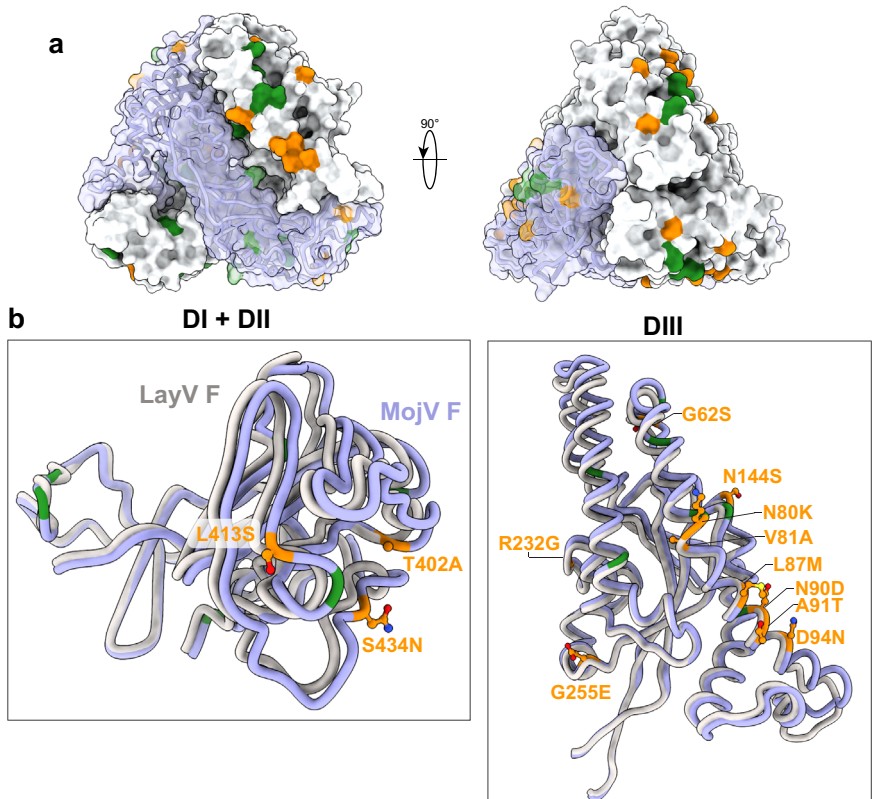

**Fig. 2 | Sequence and structural comparison between LayV and MojV F proteins. a** MojV F protein surface representation coloured in light grey with a single monomer coloured in purple. Conserved mutations between MojV and LayV F are coloured in green and non-conserved mutations are coloured in orange. **b** Model comparisons between LayV F (grey) and MojV F (purple) within different domains. Conserved mutations are coloured green and non-conserved mutations are coloured orange. Non-conserved MojV F side chains are depicted as sticks with oxygens coloured red and nitrogen coloured blue.

was transfected into ExpiCHO-S cells, which yielded ~5.3 mg/L of recombinant prefusion-stabilized F. Protein purification from cell culture supernatant was made possible by immunoaffinity with a monoclonal antibody made in-house targeting the proprietary clamp2 domain. Prefusion LayV F proteins displayed the expected monomeric molecular weight of ~66 kDa on SDS-PAGE under reducing conditions and were devoid of contaminants (Fig. 1b). LayV F was further purified to homogeneity by size-exclusion chromatography (SEC), where a major trimeric peak was observed (Fig. 1c). In contrast, unstabilized LayV F proteins displayed a SEC profile with both trimer and aggregate peaks present (Fig S1), likely due to exposure of the hydrophobic fusion peptide leading to aggregation as well as lack of stabilization leading to dissociation of $F_1$ and $F_2$ subunits. Interestingly, unstabilized MojV F prepared via an analogous method appeared trimeric on SEC, however further investigation by negative stain electron microscopy revealed a typical postfusion conformation (Fig. S1), consistent with previous reports of unstabilized HNV F proteins[39]. We found that addition of the clamp2 domain aided in stabilizing the prefusion form. Indeed, 2D class averages generated from initial cryo-EM images revealed particles resembling the canonical three-fold symmetry HNV F protein structure (Fig. S1). Concurrently, we applied the same methodology to MojV F, where ectodomain residues 1–482 were used to express the prefusion conformation (Fig. S2).

We proceeded to determine the structure of LayV F via single particle cryo-EM (Fig. 1d). We collected 2875 movies, which were curated to 2609 based on CTF fit resolution (Figs. S3, S4). Through iterative particle picking, 2D classification and 3D refinements, we obtained a final structure of LayV F at 3.37 Å (Fig. 1d, Fig. S3). In our analysis, we observed two particle populations that either had the stem domain displayed or not. The stem can be hyperflexible resulting in poor resolution around this region. Several rounds of heterogenous

refinements and ab initio reconstructions using cryoSPARC were required to remove particles where the stem was not resolved (Fig. S3). This yielded a final map including both the globular head domain and stem and resembled the "tree-like" conformation seen before for other HNV F proteins. Using this structure, we defined the different domains (DI-III) within LayV F that are typically present in HNV F proteins. LayV F DI is central to the monomer and is flanked by DII and DIII, the latter of which contains the fusion peptide (FP) (Fig. 1a, d). Heptad repeat B (HRB) resides in the membrane proximal region and constitutes part of the stem domain.

Next, we determined the structure of MojV F to 2.66 Å (Figs. S2, S5). Here, a total of ~260,000 particles were extracted and 2D classified from ~9000 curated movies (Figs. S4, S5). These were further classified using ab initio and heterogenous refinements within cryoSPARC to yield ~214,000 particles that were subsequently used in a homogenous refinement. Interestingly, while the stem domain for MojV was observed in 2D classes, this region was not observed in any 3D classification or homogenous refinements and therefore was not resolved to a high resolution (Figs. S2, S5). Structural comparisons of LayV and MojV F ectodomains revealed several surface exposed, conserved (L23I, I33V, R64K, F71Y, K79R, L89I, K197R, L206I, E267D, V298I, V328I, E330D, Y345H, R370K, K411R, I420V, E430D, V444I) and non-conserved (G62S, N80K, V81A, L87M, N90D, A91T, D94N, N144S, R232G, G255E, T402A, L413S, S434N, N468E) substitutions in DI, DII and DIII (Fig. 2). These are differences that should be noted, however we anticipate that cross-reactive responses between the two would be possible, given the high sequence similarity in the F glycoproteins (~90%).

Despite an overall high structural resemblance between NiV prefusion F and the divergent LayV (RMSD = 1.8 Å) and MojV F (RMSD = 2.6 Å) structures, there are some notable differences. Comparison of

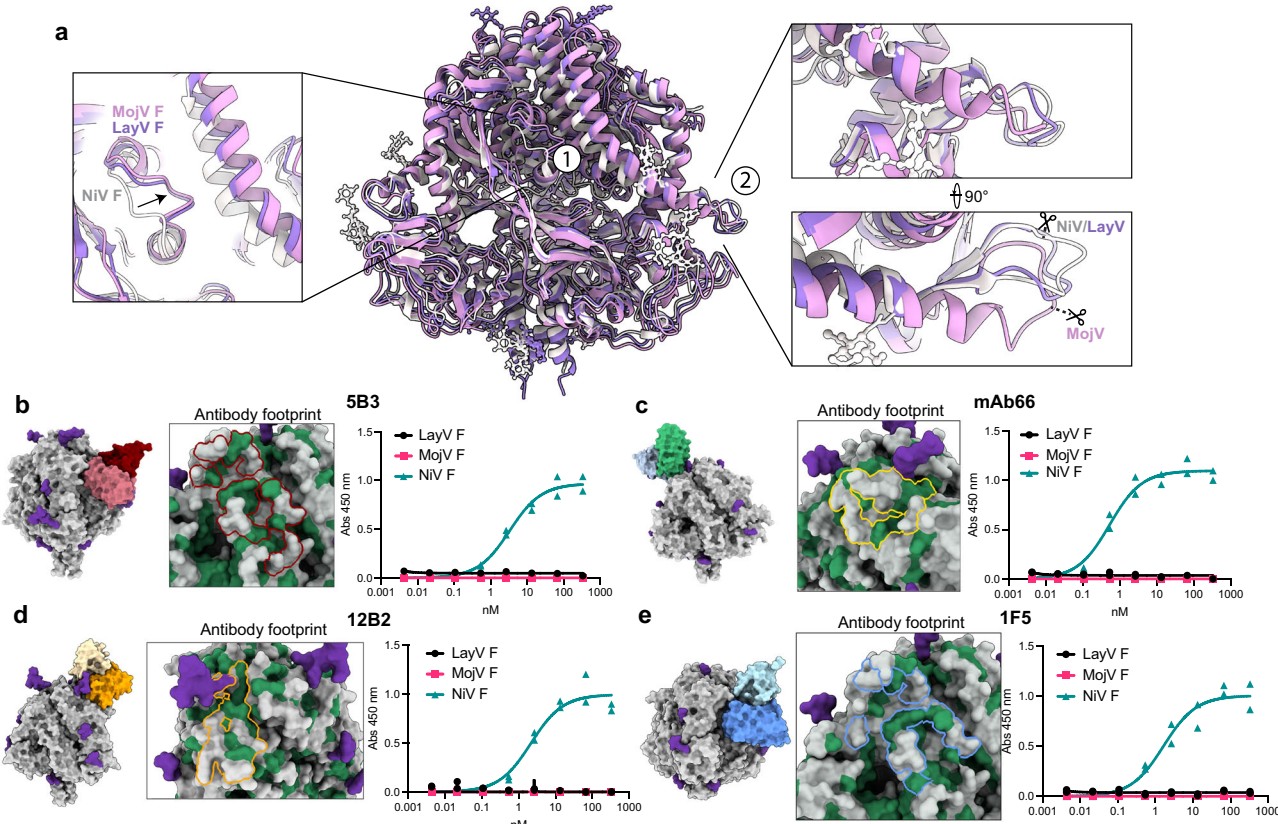

**Fig. 3 | LayV and MojV F are antigenically distinct from NiV F. a** LayV F (purple), MojV F (pink) and NiV F (white) trimer overlay. Inset 1 shows inter-protomer DIII contacts which are shared for LayV and MojV but distinct from NiV, a shift in a surface loop (arrow) projects into the interprotomer interface resulting in lateral translocation of the opposing DIII helix. Inset 2 shows structural homology between LayV and NiV at the F1/F2 cleavage site compared to MojV, cathepsin cleavage site is indicated. Antibody binding site and footprint are shown for 5B3 (**b**), mAb66 (**c**), 12B2 (**d**) and 1F5 (**e**). Indirect ELISAs of each mAb are shown to the right against prefusion clamp2 stabilized NiV F, MojV F and LayV F. Two replicates for each datapoint are shown. In each antibody footprint map, NiV F is shown with conserved residues in LayV F depicted as green and non-conserved as grey, with glycans coloured in purple. Source data are provided as a Source Data file.

the trimers reveals a shift in the major outward facing DIII helix of LayV F relative to NiV, mediated by an adjacent loop region of a neighbouring protomer, which adopts an altered conformation in both MojV and LayV (Fig. 3a inset 1). Conversely, the F1/F2 cleavage site of LayV more closely resembles the NiV and HeV site, despite high sequence homology between LayV and MojV in this region (Fig. 3a inset 2). The alternative cleavage site conformation for LayV is likely due to the presence of a bulkier hydrophobic leucine residue in position 413 in the adjacent DII (serine in the equivalent site in MojV), which would otherwise clash with the conserved and cleavage-site proximal tyrosine 102 (Y107 in MojV). The MojV and LayV F cryo-EM structures from this work also display clear fusion peptide loop densities, which were only previously seen in structures determined by X-ray crystallography for NiV (PDB 5EVM & 6T3F) (Fig. S6).

### LayV and MojV F do not bind known HNV F antibodies or react to polyclonal sera

To further explore the antigenicity and divergence of LayV and MojV, we conducted ELISA binding assays against recombinantly expressed F proteins with four known neutralizing HNV F-specific antibodies: 5B3, mAb66, 12B2 and 1F5[21–23] (Fig. 3). While these mAbs are known to bind and neutralize both NiV and HeV, we found that this does not extend to LayV or MojV. LayV and MojV F glycoproteins only share ~40% sequence identity with NiV F, which further speaks to their divergence from the canonical HNVs (Fig. S7). Indeed, structural analyses revealed that the binding sites of these mAbs are non-conserved in LayV and MojV F (Fig. 3). We also investigated whether mouse polyclonal NiV F-specific antiserum, raised through prefusion F subunit vaccine

immunization studies previously conducted[25], was capable of cross-reacting to LayV and MojV F proteins. We found that polyclonal sera raised against prefusion NiV F provided little to no cross-reactivity to MojV and LayV, carrying implications for broad-spectrum vaccine designs against these pathogens (Fig. S7).

### LayV and MojV F exhibit a differential glycosylation pattern to NiV F

Sequence analysis of LayV and MojV F proteins reveals three putative *N*-linked glycosylation sites at positions N65, N279 and N459 of LayV F, and N69, N283 and N463 of MojV F. In contrast, NiV F possesses five putative *N*-linked glycan sites, with only four occupied (N67, N99, N414 and N464) (Fig. 4)[32,40]. The LayV F cryo-EM map possessed additional electron map densities to the generated model that were positioned at N459 and N65 (Fig. 4a). Interestingly, despite being surface exposed, no glycan density was observed at position N279, suggesting that only 2 of 3 potential *N*-linked glycan sites are occupied in LayV F. To confirm these observations, we performed mass spectrometry glycoproteomic analysis of ExpiCHO-S expressed LayV F, which revealed complete occupancy at position N65 with complex bi- and tri-antennary, core fucosylated *N*-glycan structures (Fig. 4b, d). Notably, this glycan site is within a previously determined site of vulnerability for NiV, and would likely block the activity of mAb66-like antibodies for LayV and MojV[21].

Mass spectrometry glycoproteomic analysis also revealed that residue N459 is fully occupied by a mixture of oligomannose, hybrid and tri- and tetra-antennary complex glycan structures with and without core fucosylation (Fig. 4c, d). Aligning with the cryo-EM map, mass spectrometry analysis showed that there was no detectable

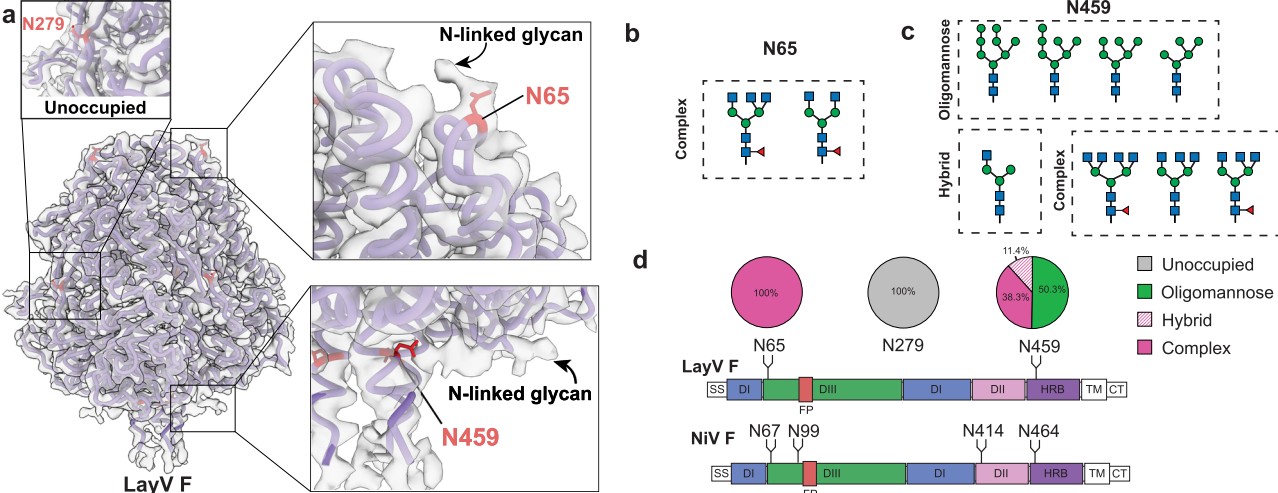

**Fig. 4 | Glycan occupancy in LayV F. a** LayV F cryo-EM map (grey) glycan identification fitted with generated model (purple). *N*-linked sites coloured red and Asn residues shown as sticks. Glycan structures observed at N65 (**b**) and N459 (**c**) by mass spectrometry glycoproteomics. *N*-Acetylglucosamine is presented as blue squares, mannose as green circles and fucose as red triangles. **d** Schematics highlighting the position of the occupied glycosylation sites in LayV and NiV F and the proportion of each glycan type at the sites in LayV.

glycosylation at position N279. Although we observed clear glycan cryo-EM volume at N459 and only weak density at N65, mass spectrometry analysis showed complete glycosylation occupancy at both sites. The glycan structures we observed at N65 were highly processed complex *N*-glycans, consistent with the high accessibility of this site. In contrast, the glycan structures at N459 were a mixture of oligomannose and highly processed hybrid and complex *N*-glycans. The relatively less-processed *N*-glycans at this site may be due to its less accessible location at the top of the stem domain, and also due to potential interactions between the glycan and amino acids in the globular head domain, as suggested by the clear glycan cryo-EM volume.

## Discussion

Over the past decade, several new HNV have been discovered, significantly expanding the viral genus beyond the prototypical NiV and HeV members[4,6–8,41]. These novel viruses were discovered either in sequence or as isolates from geographically distinct regions and from different reservoir hosts. In August 2022, a report in the New England Journal of Medicine identified a novel HNV, LayV, which caused severe pneumonic infections in a cluster of human patients in China[8]. These were the first cases of human HNV infection outside of NiV and HeV. Together, this highlights the emergence of HNVs globally and the imminent threat of spillover into human populations, with the potential of HNV outbreaks to cause significant mortality and morbidity, as seen in previous NiV outbreaks. Despite this, there are currently no approved vaccines or therapeutics for HNVs. A HeV G subunit vaccine has shown promise in preclinical development and is now progressing to human trials[10–13], however this vaccine candidate is unlikely to cross-protect against other HNVs outside of NiV and HeV. Furthermore, little is currently known about divergent HNVs such as LayV and MojV in terms of receptor usage[42], and there are limited biological tools and antibodies to facilitate further investigation. To this end, we determined the structure of the prefusion F glycoproteins of LayV and MojV to high resolution via cryo-EM and performed biochemical & biophysical analyses to probe their antigenicity.

We found that LayV F and MojV F both adopt a "tree-like" prefusion F structure similar to that of NiV, despite only sharing ~40% sequence conservation at the amino acid level. This likely reflects the nature of RNA viruses which are prone to high mutation rates, yet are subjected to similar selection pressures and therefore retain an overall

structure and architecture to maintain functionality. While the proteins resemble one another structurally, no cross-reactivity of NiV F-specific mouse antisera was observed to MojV and LayV F proteins. This carries implications for both vaccine preparedness and immunotherapy development. In particular, this suggests that a single HNV immunogen would not be effective as a broad-spectrum vaccine and that a trivalent or tetravalent approach may be required. Alternatively, putative HNV ancestral sequences can be predicted and tested as broad-spectrum vaccine candidates, a strategy previously used for HIV and influenza viruses[43–45]. Of note, the HNV F glycoproteins possess higher sequence conservation (ranging from ~32 to 90%) in comparison to HNV G (ranging from ~15 to 78%), indicating that F may be a more suitable candidate in the context of a broad-spectrum vaccine.

Several NiV and HeV F-specific antibodies have recently been characterized and shown to be both neutralizing and protective[21–24]. In this work, we found that these mAbs failed to bind LayV and MojV F proteins, and this can be explained by the low level of homology for surface-exposed residues. Further, the identification of a glycan at the DIII apex would shield LayV and MojV from mAbs such as mAb66. However, our findings show that LayV F and NiV prefusion F do share regions of conservation that may be permissible to antibody binding. One such region is a conserved pocket which encompasses DI and DII of a single protomer and DIII of a neighbouring protomer. Another semi-conserved pocket is present primarily within DI of a single protomer (Fig. S7). While these sites are conserved, they encompass a small surface area that may be more suitable for nanobody immunotherapies, which are more amenable to binding protein pockets and crevices in comparison to conventional mAbs, which require larger binding interfaces[46–48]. Moreover, such semi-conserved sites are analogous to epitopes that exist within RSV and human metapneumovirus (hMPV) F proteins, which only share ~33% in sequence identity, yet can be targeted by cross-neutralizing antibodies such as MPE8[49]. In contrast, we anticipate that little cross-reactivity between the G glycoproteins of NiV and LayV would occur given the low sequence identity (~23% at the amino acid level) and disparate receptor usage by MojV[42].

Sequence analysis of LayV F reveals three putative *N*-linked glycosylation sites at N65, N279 and N459. This is in contrast to NiV F, which has 5 putative *N*-linked glycan sites, but with only four occupied (N67, N99, N414, N464)[32,40]. Previous studies have demonstrated that the glycan at N414 is essential for cell surface expression and fusogenicity of NiV F and CedV F (present at N413 in CedV F)[32,50].

Interestingly, the MojV and LayV F sequences do not contain an N-linked glycan motif at this position. In fact, our cryo-EM structure coupled with mass spectrometry analysis revealed that only positions N65 and N459 are glycosylated on LayV F, demonstrating that this glycoprotein is significantly less glycosylated in comparison to NiV. This is highly unusual for paramyxovirus glycoproteins, given the importance of glycans in host immune evasion, protein folding and function, and virulence[32,40,50–56]. Interestingly, a similar phenomenon was seen for MojV attachment protein, which was observed to have very few glycosylation sites, yet retained its functionality[42]. As such, the lack of extensive glycosylation on LayV and MojV F and G proteins may be a distinguishing feature of these newly emergent viruses from the prototypical HNV genus. In particular, this may carry implications for vaccine and therapeutic design, as lack of glycans on the major surface glycoproteins may translate into increased vulnerability to antibody neutralization. Of note, the LayV and MojV F glycoproteins were expressed in the ExpiCHO-S system, which may not reflect the glyco-form distribution present in human cells and therefore may differ from virus isolated from human infections.

In this work, we have determined the structures of LayV and MojV F glycoproteins in their prefusion conformation via cryo-EM, per-formed occupancy and glycoform analysis of LayV F, and probed both proteins for their antigenicity with known F mAbs and sera. Our work carries implications into rational antigen design for this viral genus as well as broad-spectrum vaccine strategies. The COVID-19 pandemic has indicated that the world is underprepared in the face of viral outbreaks, and as such structural insights into the major glycoproteins of emerging human pathogens are valuable in both expanding our understanding of virus biology and in epidemic vaccine preparedness.

## Methods

### Antigen design, expression and purification

The LayV F (GenBank UUV47205.1) and MojV F (Genbank YP_009094094.1) genes were codon optimized and ordered via Inte-grated DNA Technologies (IDT) as synthetic gene blocks. Upon receipt, the F ectodomains (amino acids MojV:1-482; LayV:1-478) were amplified via PCR and cloned into pNBF mammalian expression vectors via inFusion cloning and Stellar transformation, as per manufacturer's protocol (TakaraBio). The ectodomains were cloned under a CMV promoter upstream of a proprietary molecular clamp trimerization domain (clamp2) linked by a GSG linker, analogous to that previously described[25,28,38]. Sequence confirmed clones were transiently trans-fected into ExpiCHO-S cells (ThermoFisher), as per manufacturer's instructions. Briefly, 1 µg plasmid DNA was transfected per 1 mL of ExpiCHO cells at a density of $1 \times 10^6$ cells/mL. Seven days post-trans-fections, cell culture supernatants were harvested by centrifugation at $4800 \times g$ for 30 min at 4 °C before filter sterilization (0.22 µm). Antigens were then purified using affinity chromatography on an ÄKTA Pure system (Cytiva). Here, supernatants were passed through an in-house made NHS column (Cytiva) conjugated with an antibody specific to the trimerization domain. The column was washed with 10 column volumes (CVs) of PBS and antigens were eluted using a high pH elution buffer (0.1 M glycine, 0.4 M NaCl, 5 mM EDTA, pH 11.5) and immediately neu-tralized with a 1:1 v/v ratio of 1 M Tris pH 6.8. Peak elution fractions were collected and concentrated and buffer exchanged to PBS using 30 K MWCO centrifugal concentrators (Merck Amicon).

Unstabilised LayV and MojV F proteins were expressed as above without the clamp2 trimerization domain. A C-Tag (EPEA) was added to the ectodomain C-terminus linked by a GSG linker to enable purifica-tion. Proteins were expressed in ExpiCHO-S cells as described above and purified from supernatants using CaptureSelect™ C-Tag Affinity Matrix (ThermoFisher) as per manufacturer's protocol. Briefly, filtered supernatant was passed through the C-Tag column and extensively washed with C-Tag wash buffer (100 mM Tris, 150 mM NaCl, pH 7.4) before elution with C-Tag elution buffer (2 M MgCl₂, 20 mM Tris, pH 7.4). Eluted proteins were immediately concentrated and buffer exchanged to Tris buffer (100 mM Tris, 100 mM NaCl, pH 7.4) using 30 K MWCO centrifugal concentrators (Merck Amicon).

### Antibody expression and purification

Antibody heavy chain variable (Hv) and light chain variable (Lv) domains sequences were acquired from Protein Data Bank (PDB) sequences previously published: 12B2 (PDB 7KI4), 1F5 (PDB 7KI6), 5B3 (PDB 6TYS) and mAb66 (PDB 6T3F). These sequences were cloned into previously generated human IgG1 heavy chain and light chain (kappa or lambda) vectors, as previously described[57]. Antibodies were expressed in ExpiCHO-S cells as described above, with heavy and light chains transfected at a 1:3 ratio. Secreted antibodies were purified from filter-sterilized cell culture supernatants using protein A affinity chro-matography on an ÄKTA Pure system (Cytiva). After sample loading, the protein A column was washed with 10 CVs of 25 mM Tris, 25 mM NaCl, pH 7.4. Antibodies were eluted using a low pH elution buffer (100 mM sodium citrate, 150 mM NaCl, pH 3.0). Peak fractions were neutralized with a 1:1 v/v ratio of 1.5 M Tris pH 8.8 before concentration and buffer exchange to PBS as described above.

### Size exclusion chromatography

Purified MojV and LayV F antigens were subjected to size-exclusion chromatography to ascertain molecular size and as a further purifica-tion step of trimer for cryo-EM. Briefly, 100–200 µg of protein was loaded on a 500 µl loop and injected onto a Superose 6 Increase 10/ 300GL column (Cytiva). Peak trimer fractions were collected and concentrated in 0.5 mL 30 K MWCO to the desired volume and concentration.

### Negative stain transmission electron microscopy

SEC purified antigens were deposited onto glow-discharged, carbon-coated copper grids (ProSciTech) at 10 µg/mL and stained with 1 % (w/v) uranyl acetate for 2 min. Grids were imaged at 100 keV using a Hitachi HT7700 fitted with an AXT 2kx2k CMOS camera. Image processing was performed using cryoSPARC v3.3.1.

### Cryo-EM sample preparation

Prior to sample freezing, both proteins were diluted to 0.7 mg/ml and CHAPS from the VitroEase™ buffer screening kit (stock concentration 4.9%) was added to achieve a final CHAPS concentration of 0.01225%. Plunge freezing was performed using the EMGP2 system (Leica) operating at 4 °C and 95% humidity. 2 µl of sample was applied onto glow discharged Quantifoil grids (Q1.2/1.3, 300 mesh), after 3 s pre-blot incubation time, the grid was blotted using filter paper for 10 s and then plunge frozen into liquid ethane.

### Cryo-EM data collection

Cryo-EM data were collected using CRYOARM 300 (JOEL) equipped with an in column Omega energy filter and K3 detector (Gatan), housed at the University of Queensland Centre for Microscopy and Micro-analysis. The movies were acquired in super resolution and CDS modes, with 40 frames recorded per movie at a dose of 1 e/Å² per frame (40 e/Å² total dose), a target defocus range from −2.5 to −0.5 µm, and a slit width of 20 eV. Data were collected at nominal magnifications of 60,000× and 100,000×, corresponding to calibrated pixel sizes of 0.4 Å/pixel and 0.25 Å/pixel at the specimen level, for LayV F and MojV F, respectively. 2875 movies were collected for LayV F, while a total of 10,362 movies were collected over three batches for MojV F. Software used for data collection was SerialEM v.3.1 implemented with yoneolocr[58].

### Cryo-EM data processing

All data processing was performed in cryoSPARC 3.3.1[59]. All movies were motion corrected using patch Motion Correction, where the movies were Fourier cropped by a factor of 2 following which patch

CTF estimation was performed. For LayV F, a total of 2609 movies were curated based on the estimated CTF fit resolution cutoff of 6 Å, and 631,827 particles were picked. Following multiple rounds of 2D classification, ab initio reconstruction and heterogenous refinement were used to isolate the classes with a clear stem domain. The final reconstruction was obtained using homogenous refinement with C3 symmetry imposed and yielded a map at 3.37 Å resolution. The detailed processing steps are shown in Supplementary Figs. S4, S6.

For MojV F, the three batches of data were individually processed to obtain a total of 261,696 particles. Ab initio reconstruction followed by heterogenous refinement resulted in a class with clear secondary structural features (213,754 particles). Non-uniform refinement followed by homogenous refinement on the particles contain within this class with C3 symmetry imposed yielded the final map at 2.66 Å resolution. Only a small subset of particles (16,319 particles) in the 2D classes contain a visible stem domain, and thus the final reconstruction excluded the stem region. The detailed processing steps are shown in Supplementary Figs. S5, S6.

### Cryo-EM model building and analysis
The protein atomic models were initially automatically built using ModelAngelo v.0.2.2[60] and additional protein and glycan residues were added using Coot v.0.9.8.1[61]. Refinement of the models was performed using ISOLDE v.1.3[62] and the quality of the model and fit to density was determined using Phenix v.1.20.1–4487[63]. All figures were made using UCSF Chimera v.1.17 and ChimeraX v.1.3[64]. Model generation and acquisition details for LayV and MojV F proteins are summarized in Supplementary Fig. S6.

### Enzyme-linked immunosorbent assay
HNV prefusion F antigens were coated on Nunc Maxisorp immunoplates at 2 µg/mL in PBS and incubated overnight for adsorption at 4 °C. The following day, plates were blocked with 150 µl per well of 1X KPL (Seracare) in PBST (PBS + 0.1% Tween-20) for 1 h. Blocking solution was then discarded and serially titrated mAbs in block (starting at 50 µg/mL and titrated five-fold) were added at 50 µl/well for 1 h at 37 °C. Plates were washed three times in water and a secondary HRP-linked goat anti-human antibody (Invitrogen) was diluted 1:2500 in block and 50 µl/well was added for 1 h at 37 °C. Plates were washed as before prior to being developed with 50 µL/well of TMB chromogen solution (Life Technologies) for five minutes. Substrate reactions were stopped with 25 µl/well of 1 M $H_2SO_4$ and absorbance was read at 450 nm. Data were plotted with background binding against PBS coated wells subtracted and a one-site specific model fitted on GraphPad Prism v.9.4.1.

### Mass spectrometry glycoproteomic analysis of LayV F
LayV F protein samples were prepared for LC-MS/MS glycoproteomic analysis essentially as previously described[65]. Three aliquots of 5 µg purified protein were denatured and reduced by incubation in 50 µL of 6 M guanidine-HCl, 50 mM Tris HCl buffer pH 8, and 10 mM dithiothreitol at 30 °C for 30 min. Reduced proteins were alkylated by addition of acrylamide to a final concentration of 25 mM and incubation at 30 °C for 1 h. Excess acrylamide was quenched by addition of additional dithiothreitol to a final concentration of 5 mM. Proteins were concentrated and interfering substances were removed via precipitation by the addition of four volumes of methanol/acetone (1:1 v/v) and incubation at −20 °C for 16 h. Precipitated proteins were centrifuged at 18,000 × g for 10 min, and the supernatant discarded.

The protein pellets were resuspended in 20 µL of 50 mM $NH_4HCO_3$ and digested separately with sequencing grade porcine trypsin (Sigma-Aldrich, MO, USA), bovine chymotrypsin or endoproteinase Glu-C from *Staphylococcus aureus* V8 (Roche Diagnostics GmbH, Mannheim, Germany) to provide complementary sequence coverage. The ratio of enzyme to protein used were 1:20, 1:30 and 1:10

respectively, and digests were incubated at 37 °C for 16 h. Following digestion, proteolytic enzymes were inactivated by incubation at 95 °C for 5 min followed by the addition of 1 mM phenylmethylsulphonyl fluoride (PMSF) and incubation at 25 °C for 10 min. Digests were then split into 2 equal volumes, each containing ~2.5 µg of peptides. Peptides in one of the two aliquots were deglycosylated with 125 U of Peptide-N-Glycosidase F (PNGase F) (New England BioLabs, MA, USA) and incubation at 37 °C for 16 h. Peptides were desalted and concentrated with C18 ZipTips (Millipore, MA, USA). Samples were dried and reconstituted in 0.1% FA.

Samples were analysed with a ZenoTOF 7600 mass spectrometer (SCIEX) coupled to a Microscale LC system (Waters). ~100 ng of sample was injected onto a Waters nanoEase M/Z HSS T3 C18 column (300 µm × 150 mm, 1.8 µm, 100 Å). The mobile phases were A: 0.1% FA, 1% ACN, and B: 0.1% FA, 90% ACN. Loaded peptides had an initial wash for 30 s, followed by elution using a 5 µL/min flow rate in 5–35% solution B (6 min). MS1 spectra were acquired at *m/z* 350–1800 using an accumulation time of 0.2 s. The 20 most intense precursors with signal intensity of ≥ 20 and from charge state 2–5 were selected for MS2 fragmentation using collision induced dissociation (CID) and accumulation time of 0.035 s. Dynamic exclusion was enabled after two occurrences for a duration of 6 s. Zeno trap pulsing was enabled toincrease sensitivity while maintaining resolution and mass accuracy[66–68].

The trypsin digested sample was additionally analysed with parameters targeted to identification of the large, glycosylated peptide containing site N459. ~400 ng of material was injected and the LC method was modified from above to a gradient of 5–35% solution B over 22 min. MS1 spectra were acquired as described above, however precursors selected for MS2 fragmentation were required to be >1000 Da, with an intensity > 100 and an accumulation time of 0.05 s.

Peptides were identified using Byonic (Protein Metrics, v.4.3.4), searched against a database containing the protein sequences for the Langya fusion protein, proteolytic enzymes, and PNGase F. Cleavage was specified as semi-specific at the appropriate sites for each proteolytic enzyme, allowing two missed cleavages. Precursor mass tolerance was set to 20 ppm, and fragment mass tolerance to 0.1 Da. Propionamide at cysteine was set as a fixed modification. Oxidation at methionine and deamidation of asparagine were both set as "common 1" variable modifications. A database of 309 mammalian N-glycans (Byonic) were included with glycan modifications set to "rare 1". For each peptide a total of 1 "rare" modification and 2 "common" modifications were permitted. All glycopeptide spectral matches were manually validated. The mass spectrometry glycoproteomics data have been deposited to the ProteomeXchange Consortium via the PRIDE[69] partner repository with the dataset identifier PXD039898.

### Reporting summary
Further information on research design is available in the Nature Portfolio Reporting Summary linked to this article.

## Data availability
The cryo-EM data generated in this study have been deposited in the Electron Microscopy Data Bank & Protein Data Bank databases under accession codes EMDB-29299/PDB 8FMX (LayV F) and EMDB-29300/ PDB 8FMY (MojV F). The processed mass spectrometry glycoproteomics data are available at ProteomeXchange Consortium via the PRIDE partner repository with the dataset identifier PXD039898. The source data generated in this study are provided in the Source Data file. The Nipah antibody data used in this study are available in the Protein Data Bank database under accession codes 6TYS (5B3), 6T3F (mAb66), 7KI4 (12B2) and 7KI6 (1F5). The data that support the findings of this study are available from the corresponding author upon request with a Materials Transfer Agreement. Source data are provided with this paper.

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

## Acknowledgements

The authors acknowledge the facilities, and the scientific and technical assistance, of the Australian Microscopy & Microanalysis Research Facility at the Centre for Microscopy and Microanalysis, The University of Queensland. We thank Dr Amanda Nouwens and Peter Josh at The University of Queensland, School of Chemistry and Molecular Biosciences Mass Spectrometry Facility for their assistance and expertise. We would also like to acknowledge and thank Dr Rhys Parry and the authors of 'A Zoonotic Henipavirus in Febrile Patients in China' published in New England Journal of Medicine for expedited sharing of Langya sequences. N.M., Y.S.L. and A.I. would also like to thank the cryoSPARC discussion forum. We would also like to acknowledge Prof Paul Young for his ongoing support of an exceptional research environment. DW is supported by a CSL Centenary Fellowship. N.M. is supported by a Discovery Early Career Research (DECRA). B.L.S. and C.L.P. are supported by NHMRC Ideas grant (APP1186699). This work was supported by a NHMRC MRFF Coronavirus Research Response grant APP1202445 to D.W. and K.J.C. We would like to acknowledge and thank the Australian Infectious Disease Research Centre for supporting this work.

## Author contributions

D.W. and A.I. conceived and designed the study. A.I. performed antigen design. A.I. and S.T.M.C. performed antigen cloning. A.I. and B.L. performed antigen expression and validation. Y.S.L., N.M. and A.I. performed initial screening of samples for cryo-EM. Y.S.L. and N.M. performed large scale cryo-EM acquisition of datasets. A.I. and Y.S.L. performed cryo-EM dataset analysis for LayV F with guidance from D.W. N.M., Y.S.L. and A.I. performed cryo-EM dataset analysis for MojV F. A.I. and Y.S.L. performed model building. K.L.M. and J.S. performed mass spectrometry glycoproteomic analysis. B.L.S. and C.L.P. guided mass spectrometry methods and contributed to data analysis. C.A.P.S. optimised antibody purification of clamp2 constructs. B.L.S., M.J.L., K.J.C. and D.W. contributed funding for this research. A.I. wrote the manuscript with editing contributions from all authors.

## Competing interests

K.J.C. and D.W. are inventors of the 'Molecular Clamp' patent, US 2020/0040042 & PCT/IB2023/053263. The remaining authors declare no competing interests.
