## [Peer review file · Nature Communications]

Structure and antigenicity of divergent Henipavirus fusion glycoproteinsREVIEWER COMMENTS

Reviewer #1 (Remarks to the Author):

Isaacs, Low and colleagues report here on the structures of the fusion (F) glycoprotein from the two most recently discovered Henipaviruses – Mojiang (MojV) and Langya (LayV). The structures show that the two F proteins adopt a prototypical HNV fold, similar to that of Nipah – despite a low amino-acid conservation - with some discrete, but potentially impactful, local differences such as the protease activation site F1/F2. The authors complement the structural studies with immunoreactivity against polyclonal sera as well as mass spectrometry-based N-linked glycan characterization of the two F glycoproteins.

Overall, the manuscript is very straightforward, and the structures will certainly prove to be useful for any future vaccine immunogen design work.

However, the following comments should be addressed prior to publication:

- 1 – The methods section as it is now does not allow for anyone to reproduce these data and this section needs to be much more detailed. For ex.
 - the sequence of the clamp2 domain is not reported.
 - an in-house anti-clamp2 mAb was used for purification – which mAb? Please report and deposit the sequence.
 - Polyclonal sera raised against NiV F are used to test for cross-reactivity with MojV and LayV F. How were the sera raised? In what organism? What was used as immunogen? purified protein? Inactivated virus?
 - Cryo-EM sample preparation: it's unlikely that CHAPS was used at 0.255% final concentration. Please provide more detail.
 - Cryo-EM model validation: please provide the Ramachandran Z-scores
- 2 – Using select anti-NiV F monoclonal antibodies and polyclonal sera (which ones? See comment above), the authors show no cross-reactivity with MojV and LayV F. However, this does not mean that there is absolutely no cross-reactivity or cross-protection with Nipah, as anti-G immunity may prove to be more cross-protective. Could the authors test polyclonal sera from NiV survivors in pseudovirus neutralization assays complemented with reactivity by ELISA?
- 3 – The glycan composition analysis is done by MS, but the recombinant proteins were made in CHO cells, and may not reflect the N-linked glycans made in human cells, for example. This point should be emphasized throughout the manuscript.
- 4 – The authors use “electron density” to describe cryo-EM volume. This is incorrect and should be corrected throughout. Likewise, the structures are not “solved” but determined.

Reviewer #2 (Remarks to the Author):

The manuscript by Isaacs and colleagues describes the cryoEM structures for the henipaviruses Langya (LayV) and Mojiang (MojV) virus fusion proteins (F). The prefusion structures are reported at a resolution of 2.66 (MojV F) and 3.37 Å (LayV F). While LayV and MojV are members of the henipavirus genus, they are rodent-borne viruses, compared to other members in the genus, such as Nipah and Hendra viruses which are bat-borne viruses. Interestingly, the authors show that despite an only 40% sequence homology, the overall structure of LayV and MojV F is similar to the NiV F prefusion structure. However, using NiV F-specific neutralizing antibodies, no cross-reactivity could be demonstrated. This was explained through the identification of a low homology for surface exposed residues and the presence of a glycan shield in domain III for LayV and MojV F.

Overall, this is a very straight forward and timely manuscript, especially for the recently identified LayV. Methodology, data collection and data analysis are clearly described and presented, and the conclusions are according to the structural characterization. The authors conclude that the provided structures are supportive for structure-guided vaccine development and explain why a single henipavirus immunogen would likely not be effective against bat- and rodent-borne members. These data are of great interest to the henipavirus/paramyxovirus community, as well as groups focusing on structure-based development of treatment options. Future work using X-ray crystallography and solving the co-structures of F and antibodies will be important.

There are only a few minor comments that should be addressed:

lines 50-51 and line 199: Please specify that no vaccines are approved for human use. A veterinary HeV G subunit vaccine is available. Also, please add references 34-36, as has been done in line 200.

Line 69: Please define "S2P" as prefusion-stabilized two-proline spikeprotein antigen.

Line 240-241: Please correct abbreviation for Cedar Virus to CedV and introduce abbreviation in line 37.

Reviewer #3 (Remarks to the Author):

In this manuscript, the authors use cryo-EM and mass spectrometry-based analysis to determine and compare the glyco-profile of the fusion protein of Langya virus with Nipah and Hendra viruses. Overall, the manuscript is well written but lacks coherent information on the glycosylation of the viral fusion glycoproteins. Please address the comments below before the work can be accepted for publication.

1. Given that the viral proteins are largely glycosylated, the introduction need to have a short section on viral glycosylation (N-/O-types, relevance of glycosylation on biological functions, analytical methods to determine, existing literature that compare the glycosylation patterns between different HNV viruses etc).
2. L76, the objectives include solving structures using cryo-EM, but what about the MS-based approaches? Wasn't that also a complementary technique used to determine the glycosylation of LayV/MojV etc?
3. Some background information/relevance of D-domains would be beneficial for readers.
4. L165 – since the glyco-sites have not been introduced until here, the unoccupied site should also be mentioned here.
5. Are all the sites for LayV/MojV F-protein 100% glycosylated?
6. Fig 4C is showing no glycans other than oligomannose – please provide more details
7. What is minimally processed oligomannose – do you mean paucimannose?
8. It is a bit unclear how the interactions between mabs and F-proteins were tested – please elaborate this section? Was there any specific cell surface glycan type that played biological roles?
9. L378 – was MS-analysis done only on LayV F or also on MojV F? Please clarify. Was the focus primarily using NMR or MS as well? This needs to be clarified in the abstract and objectives of the work.
10. Please explain why acrylamide was used for alkylation reaction? Any reason why acetone precipitation was performed after reduction/alkylation of proteins?
11. Enzymatic digestions were done sequentially? How was the data generated using multiple enzymes used in identifying the different glycoforms? Results have no mention on the utility of the multiple enzyme reaction.
12. Given that these viral proteins are heavily glycosylated, very little emphasis has been made on the glycosylation profile both throughout the manuscript. I would like to see more specific findings and details on the site occupancy and micro-heterogeneity of the fusion glycoprotein.
13. Please add details on MS parameters – resolution, ion injection time etc?

14. Unclear what different parameters were used to assess large glycosylated peptide at N459?
15. Please also highlight how the data refinement was done from Byonic outputs?

REVIEWER COMMENTS

Reviewer #1 (Remarks to the Author):

Isaacs, Low and colleagues report here on the structures of the fusion (F) glycoprotein from the two most recently discovered Henipaviruses – Mojiang (MojV) and Langya (LayV). The structures show that the two F proteins adopt a prototypical HNV fold, similar to that of Nipah – despite a low amino-acid conservation - with some discrete, but potentially impactful, local differences such as the protease activation site F1/F2. The authors complement the structural studies with immunoreactivity against polyclonal sera as well as mass spectrometry-based N-linked glycan characterization of the two F glycoproteins.

Overall, the manuscript is very straightforward, and the structures will certainly prove to be useful for any future vaccine immunogen design work.

However, the following comments should be addressed prior to publication:

1 – The methods section as it is now does not allow for anyone to reproduce these data and this section needs to be much more detailed. For ex.
- the sequence of the clamp2 domain is not reported. An in-house anti-clamp2 mAb was used for purification – which mAb? Please report and deposit the sequence.

We have now amended the details around the results (lines 109-111) & methodology (304-305). The sequence of the clamp2 trimerization domain is described in PCT/IB2023/053263 and will be publicly available 6 months after 31 Mar 2023. However, it is also possible to use the original molecular clamp sequence which has been previously been disclosed (see reference: <https://doi.org/10.3389/fimmu.2022.963023>). This includes the details of the anti-clamp antibody used to purify the original molecular clamp.

- Polyclonal sera raised against NivF are used to test for cross-reactivity with MojV and LayV F. How were the sera raised? In what organism? What was used as immunogen? purified protein? Inactivated virus?

As per the reviewer's advice, we have added additional details in line 182-184 detailing the origin of the polyclonal sera.

“We also investigated whether mouse polyclonal NiV F-specific antiserum, raised through pre-fusion F subunit vaccine immunization studies previously conducted (25), was capable of cross-reacting to LayV and MojV F proteins.”

- Cryo-EM sample preparation: it's unlikely that CHAPS was used at 0.255% final concentration. Please provide more detail.

We thank the reviewer for spotting this error and have correct the CHAPS final percentage in line 326 to 0.01225% and included the following details in lines 356-362:

“Prior to sample freezing, both proteins were diluted to 0.7 mg/ml and CHAPS from the VitroEase™ buffer screening kit (stock concentration 4.9%) was added to achieve a final CHAPS concentration of 0.01225%. Plunge freezing was performed using the EMGP2

system (Leica) operating at 4 °C and 95 % humidity. 2 µl of sample was applied onto glow discharged Quantifoil grids (Q1.2/1.3, 300 mesh), after 3 seconds pre-blot incubation time, the grid was blotted using filter paper for 10 seconds and then plunge frozen into liquid ethane.”

- Cryo-EM model validation: please provide the Ramachandran Z-scores

The Ramachandran Z score have been added to the Supplementary file S4.

2 – Using select anti-NiV F monoclonal antibodies and polyclonal sera (which ones? See comment above), the authors show no cross-reactivity with MojV and LayV F. However, this does not mean that there is absolutely no cross-reactivity or cross-protection with Nipah, as anti-G immunity may prove to be more cross-protective. Could the authors test polyclonal sera from NiV survivors in pseudovirus neutralization assays complemented with reactivity by ELISA?

In this work, we have shown that NiV F specific mAbs and mouse polyclonal sera do not cross-react to MojV or LayV F. NiV F and LayV/MojV F share ~40% sequence identity (Figure S7). By comparison, NiV G and LayV/MojV G share ~23% sequence identity. Moreover, a recent publication has demonstrated that MojV G does not bind the same receptor as NiV (Ephrin B2/B3) (see doi.org/10.1038/ncomms16060). Given this, it is unlikely that cross-reactivity exists between the G glycoproteins of these viruses. To clarify this, we have added lines 263-266 in the discussion:

“In contrast, we anticipate that little cross-reactivity between the G glycoproteins of NiV and LayV would occur given the low sequence identity (~23% at the amino acid level) and disparate receptor usage by MojV (42).”

We agree with the reviewer that pseudovirus neutralisation assays with NiV survivor serum samples would be highly interesting. However, we currently do not have access to survivor sera.

3 – The glycan composition analysis is done by MS, but the recombinant proteins were made in CHO cells, and may not reflect the N-linked glycans made in human cells, for example. This point should be emphasized throughout the manuscript.

We agree with the reviewer that this point is important to emphasize. We have added these points in the discussion (lines 283-286) and in the results (line 198).

“Of note, the LayV and MojV F glycoproteins were expressed in the ExpiCHO-S system, which may not reflect the glycoform distribution present in human cells and therefore may differ from virus isolated from human infections.”

“To confirm these observations, we performed mass spectrometry glycoproteomic analysis of ExpiCHO-S expressed LayV F, which revealed complete occupancy at position N65 with complex bi- and tri-antennary, core fucosylated *N*-glycan structures (Figure 4B and 4D)”

4 – The authors use “electron density” to describe cryo-EM volume. This is incorrect and should be corrected throughout. Likewise, the structures are not “solved” but determined.

We have changed the terms throughout the text, as per the reviewer’s suggestion.

Reviewer #2 (Remarks to the Author):

The manuscript by Isaacs and colleagues describes the cryoEM structures for the henipaviruses Langya (LayV) and Mojiang (MojV) virus fusion proteins (F). The prefusion structures are reported at a resolution of 2.66 (MojV F) and 3.37 Å (LayV F). While LayV and MojV are members of the henipavirus genus, they are rodent-borne viruses, compared to other members in the genus, such as Nipah and Hendra viruses which are bat-borne viruses. Interestingly, the authors show that despite an only 40% sequence homology, the overall structure of LayV and MojV F is similar to the NiV F prefusion structure. However, using NiV F-specific neutralizing antibodies, no cross-reactivity could be demonstrated. This was explained through the identification of a low homology for surface exposed residues and the presence of a glycan shield in domain III for LayV and MojV F.

Overall, this is a very straight forward and timely manuscript, especially for the recently identified LayV. Methodology, data collection and data analysis are clearly described and presented, and the conclusions are according to the structural characterization. The authors conclude that the provided structures are supportive for structure-guided vaccine development and explain why a single henipavirus immunogen would likely not be effective against bat- and rodent-borne members. These data are of great interest to the henipavirus/paramyxovirus community, as well as groups focusing on structure-based development of treatment options. Future work using X-ray crystallography and solving the co-structures of F and antibodies will be important.

There are only a few minor comments that should be addressed:

lines 50-51 and line 199: Please specify that no vaccines are approved for human use. A veterinary HeV G subunit vaccine is available. Also, please add references 34-36, as has been done in line 200.

We have corrected the introduction as per the reviewer’s suggestion and added the following for the reader (lines 56-58):

“There are currently no approved vaccines for HNVs, however a HeV G vaccine is available for veterinary use and an analogous subunit vaccine is currently in clinical development for human use (10-13).”

Line 69: Please define “S2P” as prefusion-stabilized two-proline spike protein antigen.

Line 240-241: Please correct abbreviation for Cedar Virus to CedV and introduce abbreviation in line 37.

We thank the reviewer for their comments and corrected the manuscript as per their suggestions.

Reviewer #3 (Remarks to the Author):

In this manuscript, the authors use cryo-EM and mass spectrometry-based analysis to determine and compare the glyco-profile of the fusion protein of Langya virus with Nipah and Hendra viruses. Overall, the manuscript is well written but lacks coherent information on the glycosylation of the viral fusion glycoproteins. Please address the comments below before the work can be accepted for publication.

1. Given that the viral proteins are largely glycosylated, the introduction need to have a short section on viral glycosylation (N-/O-types, relevance of glycosylation on biological functions, analytical methods to determine, existing literature that compare the glycosylation patterns between different HNV viruses etc).

The following text has been included in the introduction (lines 79-89):

“The F and G proteins of HNV are glycoproteins, as they are post-translationally modified with oligosaccharide structures known as glycans. Two common types of glycans, N-linked or O-linked, are covalently attached to nitrogen or oxygen atoms in amino acid side chains, respectively. Protein glycosylation can alter immunity and antibody sensitivity by shielding or exposing viral protein epitopes, as has been observed for SARS-CoV-2 (29) and HIV (30). Glycosylation can also impact the structures and dynamics of proteins. With respect to HNV F proteins, mutations of the sites of N-glycosylation have shown they are required for proper folding and processing of NiV and HeV F (31, 32). Site-specific glycosylation of glycoproteins can be studied using mass spectrometry glycoproteomics, as has been applied to HeV G (31). However, there are few studies investigating the glycosylation of HNV F proteins, which is likely to be important in the context of immunity and vaccine design”

2. L76, the objectives include solving structures using cryo-EM, but what about the MS-based approaches? Wasn't that also a complementary technique used to determine the glycosylation of LayV/MojV etc?

We have revised the text here to include glycosylation analysis in the objectives (line 99-103):

“Here, we use a similar approach to determine the structures of LayV and MojV F glycoproteins in the prefusion conformation via cryogenic electron microscopy (cryo-EM) and to characterize their glycosylation profiles with mass spectrometry glycoproteomics, in order to inform future vaccine design and therapy against these emerging viruses.”

3. Some background information/relevance of D-domains would be beneficial for readers.

We have provided background information and relevance of the HNV F domains in the introductions, lines 62-72:

“The HNV F protein is a trimeric type I fusion protein that is initially expressed as an F precursor (F₀), consisting of three domains (DI, DII and DIII), a C-terminal stem, a transmembrane domain and a cytoplasmic tail. During infection, the F₀ protein is cleaved by cathepsin L protease into F₁ and F₂ subunits, which are linked by disulfide bonds and together constitute the fusogenic F protein (14-16). The prefusion conformation of NiV and HeV F have been determined along with the fusion core of NiV (17-19). These structures

highlight the large conformational changes that take place within F during fusion, where two heptad repeats in DIII and the stem (HRA and HRB) coalesce into a six-helix bundle, which is required for insertion of the fusion peptide into the host cell membrane (20). This outlines a rational basis for neutralizing antibody discovery and structure-based vaccine design against prefusion F.

4. L165 – since the glyco-sites have not been introduced until here, the unoccupied site should also be mentioned here.

The glycosylation analysis described in this paragraph and the next fully describes the glycosylation occupancy and structures at all of these sites.

5. Are all the sites for LayV/MojV F-protein 100% glycosylated?

Two of the three sites of LayV are fully occupied, as described in the results section. We have also cross referenced Figure 4D at the relevant position in the results section. To clarify, we have also revised the text of the Figure legend for 4D.

“(D) Schematics highlighting the position of the occupied glycosylation sites in LayV and NiV F and the proportion of each glycan type at the sites in LayV.”

6. Fig 4C is showing no glycans other than oligomannose – please provide more details

Three glycan types (oligomannose, hybrid and complex) are depicted in Fig. 4C.

7. What is minimally processed oligomannose – do you mean paucimannose?

This refers to oligomannose that have not undergone substantial trimming in the *cis*-Golgi. We agree the term may be confusing and have removed the term “minimally processed” completely from the sentence.

“In contrast, the glycan structures at N459 were a mixture of oligomannose and highly processed hybrid and complex N-glycans”.

8. It is a bit unclear how the interactions between mAbs and F-proteins were tested – please elaborate this section? Was there any specific cell surface glycan type that played biological roles?

Interactions between mAbs and F proteins were tested using an ELISA assay against recombinantly expressed proteins. We have highlighted this in lines 175-177:

“To further explore the antigenicity and divergence of LayV and MojV, we conducted ELISA binding assays against recombinantly expressed F proteins with four known neutralizing HNV F-specific antibodies: 5B3, mAb66, 12B2 and 1F5 (21-23) (Figure 3).”

The methods for this are also outlined in the methods section (under ELISA). None of the mAbs tested (5B3, 1F5, 12B2 or mAb66) interact with cell surface glycans on NiV F.

9. L378 – was MS-analysis done only on LayV F or also on MojV F? Please clarify. Was the focus primarily using NMR or MS as well? This needs to be clarified in the abstract and objectives of the work.

Only LayV F protein was analysed by MS glycoproteomics. We have revised the text in line 419 to clarify this point:

“LayV F protein samples were prepared for LC-MS/MS glycoproteomic analysis”

10. Please explain why acrylamide was used for alkylation reaction? Any reason why acetone precipitation was performed after reduction/alkylation of proteins?

Acrylamide was used as an alkylating agent to reduce the incidence of side reactions. The following reference (10.1074/mcp.M116.064048) compares alkylating agents and found acrylamide was superior to iodine containing reagents.

Acetone precipitation was performed after denaturation, reduction and alkylation to concentrate the proteins and remove contaminants. This has been noted in the relevant section in lines 424-426:

“Proteins were concentrated and interfering substances were removed by precipitation through the addition of four volumes of methanol/acetone (1:1 v/v) and incubation at -20 °C for 16 h”.

11. Enzymatic digestions were done sequentially? How was the data generated using multiple enzymes used in identifying the different glycoforms? Results have no mention on the utility of the multiple enzyme reaction.

The enzyme digests were performed separately. The following sentence was amended to clarify this for the reader in line 429:

“The protein pellets were resuspended in 20 µL of 50 mM NH₄HCO₃ and digested separately with sequencing grade porcine trypsin (Sigma-Aldrich, MO, USA), bovine chymotrypsin or endoproteinase Glu-C from Staphylococcus aureus V8 (Roche Diagnostics GmbH, Mannheim, Germany) to provide complementary sequence coverage.

12. Given that these viral proteins are heavily glycosylated, very little emphasis has been made on the glycosylation profile both throughout the manuscript. I would like to see more specific findings and details on the site occupancy and micro-heterogeneity of the fusion glycoprotein.

A key result of our study is rather that LayV F and MojV F are substantially less glycosylated than their HeV and NiV counterparts. Clear and complete descriptions of the site occupancy and glycoform distributions at each site in LayV F is reported in the results and in Figure 4. We note that we have also revised the text to include a more detailed introduction to glycosylation of viruses in the introduction (see response to Reviewer 3 Comment 1). We have also revised the Discussion text (lines 288-290):

“In this work, we have solved the structures of LayV and MojV F glycoproteins in their prefusion conformation via cryo-EM, performed occupancy and glycoform analysis of LayV, and probed both proteins for their antigenicity with known F mAbs and sera”.

13. Please add details on MS parameters – resolution, ion injection time etc?

The relevant accumulation times have been included in the MS parameter sections. TOF resolution is dependent on the flight path of the instrument and is not a selectable parameter. To provide clarity about instrument capabilities we have included the following sentence (lines 450-451) “*Zeno trap pulsing was enabled to increase sensitivity while maintaining resolution and mass accuracy*” and included three citations for the ZenoTOF prototypes that describe resolution capabilities.

14. Unclear what different parameters were used to assess large glycosylated peptide at N459?

This section has been altered to include more detail and clearly reference that the base parameters used (lines 454-458):

“~400 ng of material was injected and the LC method was modified from above to a gradient of 5-35% solution B in 22 min. MS1 spectra were acquired as described above, however precursors selected for MS2 fragmentation were required to be > 1000 Da, with an intensity > 100 and an accumulation time of 0.05 s.”

15. Please also highlight how the data refinement was done from Byonic outputs?

The following sentence has been included in line 468: “All glycopeptide spectral matches were manually validated.”